# Completeness of obstetric referral letters/notes from subdistrict to district level in three rural districts in Greater Accra region of Ghana: an implementation research using mixed methods

Mary Amoakoh-Coleman,[1,2] Evelyn Ansah,[3] Kerstin Klipstein-Grobusch,[2,4] Daniel Arhinful[1]

For numbered affiliations see end of article.

**Correspondence to**
Mary Amoakoh-Coleman;
menba19@yahoo.com

## ABSTRACT

**Objective** To assess the completeness of obstetric referral letters/notes at the district level of healthcare.

**Design** An implementation research within three districts in Greater Accra region using mixed methods. During baseline and intervention phases, referral processes for all obstetric referrals from lower level facilities seen at the district hospitals were documented including indications for referrals, availability and completeness of referral notes/forms. An assessment of before and after intervention availability and completeness of referral forms was carried out. Focus group discussions, non-participant observations and in-depth interviews with health workers and pregnant women were conducted for qualitative data.

**Setting** Three (3) districts in the Greater Accra region of Ghana.

**Participants** Pregnant women referred from lower levels of care to and seen at the district hospital, health workers within the three districts and pregnant women attending antenatal clinic in the district and their family members or spouses.

**Intervention** An enhanced interfacility referral communication system consisting of training, provision of communication tools for facilities, formation of hospital referral teams and strengthening feedback mechanisms.

**Outcome** Completeness of obstetric referral letters/notes.

**Results** Proportion of obstetric referrals with referral notes improved from 27.2% to 44.3% from the baseline to intervention period. Mean completeness (95% CI) of all forms was 71.3% (64.1% to 78.5%) for the study period, improving from 70.7% (60.4% to 80.9%) to 71.9% (61.1% to 82.7%) from baseline to intervention periods. Health workers reported they do not always provide referral notes and that most referral notes are not completely filled due to various reasons.

**Conclusions** Most obstetric referrals did not have referral notes. The few notes provided were not completely filled. Interventions such as training of health workers, regular review of referral processes and use of electronic records can help improve both the provision of and completeness of the referral notes.

## Strengths and limitations of this study

► Use of both quantitative and qualitative methods allowed us to both triangulate and explain the findings from the perspective of the health worker who refers patients and writes the letters/notes.

► Assessment of referral letters/notes at the referral hospital only (outside the referring facility) did not allow us to assess referring provider and contextual factors associated with the completeness of the referral notes.

► Also as evident from the results of the qualitative data, it is possible some participants were given referral notes but did not present them at the referral facility.

► Available resources allowed us to implement and evaluate the intervention package only for a relatively short period, with possibility of limiting the impact of the intervention.

## INTRODUCTION

The unpredictability and urgency of most obstetric complications and emergencies that require referrals demands that delays are avoided. Maternal referrals are unavoidable due to unequitable distribution of healthcare resources. Support systems like effective communication are important during obstetric referrals, as they facilitate the needed emergency care process and reduce barriers of distance and time.[1] Also, the quality of care for referred patients and referral feedback mechanisms are enhanced when there is an initial direct contact between the referring and receiving physicians.[2 3]

Communicating patient information at the time of referral is important for high-quality care and outcome, and care givers at higher levels of care value this information exchange for shared patients.[4] Several

problems have been identified regarding effective communication by health providers during referrals. These relate to specifying the main reason for and result of consultation, inadequately written medical reports and unclear follow-up plans among others.[5–9] The absence of this shared information creates dissatisfaction among providers of care. The reasons for dissatisfaction include delayed or missing referral letter, missing information in the referral communication, time required to write a referral note and difficulty in finding a specialist.[5 6 8–12] It has been acknowledged that effective communication around referrals facilitates processes needed for referral, including transportation.[1] Interfacility communication makes it possible for the referring facility to confirm that the referral facility has the needed services, provider and logistics for the patient at the time of referral. The referral centre is able to adequately prepare to receive patients when they are informed about the patient ahead of her arrival. This helps to avoid waste of time resulting from referred patients moving from one place to another for the needed care. The referral letter or note serves as a useful communication tool for referrals. However, often inter-facility communication is limited, and written notes offer limited information for patient care because of their quality.[13] A review of surgical referrals in a tertiary hospital in Ghana showed incomplete referral forms for all participants, with more missing essential items when structured referral forms are not used compared with when they are used.[14]

In Ghana, at the district level of the primary healthcare, obstetric referrals are from lower levels such as community-based health planning and services (CHPS) compounds, community clinics and health centres to the district hospital. On referral of patients for any condition including obstetric care, a referral note or letter has to be written using a referral form. The filled referral form describes who the patient is, his or her complaints, general and obstetric examination findings and laboratory investigations, diagnosis, what treatment has been given or started, reason for referral and contact of the referring provider.[15] Limited work has been done on quality of obstetric referrals and specifically on the quality of obstetric referral notes in the Ghanaian context.[16] Our aim was to assess the completeness of the referral letters/notes that pregnant women are given when referred from the subdistrict level to the district level for care.

## METHODS
### Design and setting
This study is part of an implementation research to evaluate the role of an enhanced interfacility communication system on the processes and outcomes of maternal referrals in three districts/municipalities in the Greater Accra region of Ghana from May 2017 to January 2018. It employed a mixed methods approach. The qualitative methods were to enable us interrogate potential explanations behind some of the quantitative findings. Quantitative assessment was undertaken by surveys involving the use of a before and after design, while for the qualitative assessment, focus group discussions (FGDs), non-participant observations and in-depth interviews (IDIs) were conducted. A composite intervention package of an enhanced interfacility communication system was put in place and run for 4 months after 4 months of baseline data collection.

The Greater Accra region hosts Ghana's capital city and has 20 administrative metropolises, municipalities, districts and submetropolises. It is mostly urban but has four rural districts. Available resources for this work did not allow us to work in the purely urban districts that have a more complex network of referrals from both public and private facilities. The districts we worked in are districts A, B and C (pseudo-names used for anonymity and confidentiality) and are largely rural[2] or periurban, with a higher population.[1] Two of the selected districts (districts A and B) have district public hospitals, while one (district C) has a polyclinic as the highest level public facility. It, however, has a private hospital where patients are referred to, although some patients in this district also get referred to a neighbouring district hospital that is also in another region. The different types of districts with respect to levels of care were used in this study to enable us explore the dynamics in the referral processes for the different types of services available in the district and possible implications for outcomes of care.

### The referral form
The Ghana Health Service (GHS) has, as part of its quality control measures in clinical care, designed a standard referral form that describes information that is needed to be passed on to the receiving facility about each referral. This is supplied to all facilities on request through the medical stores. It comes in duplicate in a booklet, allowing the client to be accompanied with one while the duplicate is kept in the facility for future reference. The referral form (table 1) is used for referrals for all conditions, including obstetric care, within the GHS, and also during referrals out of the GHS facilities to other facilities in the Ministry of Health (private, quasigovernment and tertiary levels). Each woman during antenatal care (ANC) receives the maternal health record book in which all record concerning the pregnancy, from antenatal through delivery to postnatal care is to be documented. During a referral, a referral form is filled. The variables on the standard form have been presented in table 1 below.

### Procedure: quantitative
A facility audit was conducted for every participating facility in the three districts to ascertain the capacity of the facility to handle referrals with respect to human resources, logistics and supplies, training, protocols and guidelines, referral forms and other related documents. Again, for every participant referred from the primary level facility to and arriving at the district hospital during

**Table 1** Variables on the referral form to be completed for referred clients

| Health facility information | Patient identification | Patient clinical information | Referring officer identification |
|---|---|---|---|
| Date | Registration number | Presenting complaints | Name of officer referring |
| Name and address of referring facility | Name | Examination findings | Position |
| Name and address of facility referred to | Sex | Temperature | Signature |
| Time referred | Date of birth | Pulse | Date/ Stamp |
| Time of departure (if emergency) | Age | Respiratory rate | |
| | Insurance status | Blood pressure | |
| | Name and address of contact person | Weight | |
| | Phone number of contact person | Results of investigations | |
| | | Diagnosis (es) | |
| | | Medical treatment/management given | |
| | | Reason for referral and comment to next level | |

the study period, we ascertained whether he or she was given a referral form. For those who had a referral form, the details of the form were captured with respect to the completeness of filling the form. For every variable that was completed on the form, a yes (1) was assigned and a no (0) when the variable is not completed.

### Procedure: qualitative

FGDs and IDIs were conducted for insight into the processes and outcomes of maternal referrals. There were six FGDs at baseline, two sets in each district, one for health workers and another for pregnant women and their spouses/partners/mothers. There were three sets after the intervention period, one for each district for health workers since they directly benefited from the intervention. Average number of participants per each FGD was 12. District, hospital and obstetric/maternity unit heads and managers provided IDIs at baseline and after the intervention period. Data collected qualitatively included information on indications for referrals, use of referrals notes and completeness of filling them, preparing clients for referrals including giving first aid, the availability and role of interfacility communication, challenges with referrals, transportation, logistics for referrals, cost of referrals to client and providers and clients' perception about referrals. Weekly non-participant observations to describe nature of majority of referrals coming into the three hospitals were done using a checklist. Discussions and interviews were conducted in English, Ga and Twi.

Research assistants were trained on data collections tools and processes. All data collections tools were pretested in a district with similar characteristics before finalised for use.

### Intervention

This is an intervention package that ensures suitability and ownership, arrived at through an assessment and understanding of what currently exists and its challenges and how this could be improved pragmatically. The design of the intervention was developed by a team comprising the study coinvestigators and a communication expert. The team reviewed and considered existing policy and relevant documents as well as previous and ongoing interventions on the subject of referrals and maternal services from relevant agencies in the GHS. The final intervention package was guided by what will be feasible and sustainable for the facilities to possibly adopt after this study.

The intervention package consisted of the following activities, and figure 1 shows how this was eventually implemented:

1. Training of health workers on interfacility referral communication including accurate documentation and use of referral notes.
2. Sharing patient information between referring and referral facilities on all referrals.
3. Provision of communication tools such as working phones and call credits for health workers to facilitate calls and text messaging.
4. Designating the task of interfacility referral communication to someone or a team in the referral facilities (including the specialist in the referral facility) and linking all such agents or teams to all the facilities within a district. These teams had monthly meetings to review maternal referrals.
5. Strengthening and enforcement of feedback mechanisms between referring and referral facilities. This includes monthly SMS reminders to referring facilities and also onsite visits to these facilities to discuss

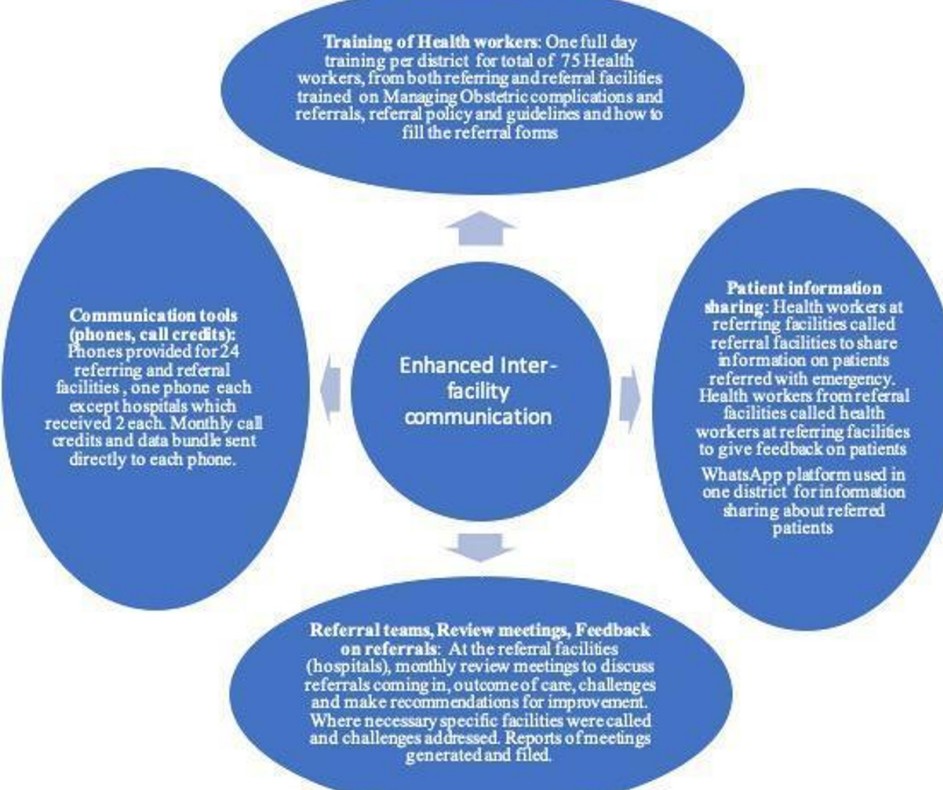

**Figure 1** Diagrammatic representation of detailed intervention roll-out.

previous referrals and provide feedback by referral teams.

## Statistical analysis

Quantitative data was entered into and analysed using IBM SPSS Statistics for Windows, V.20.0. We estimated the proportions of referred patients who were accompanied with a referral form and those for whom the standard referral forms were used. The percentage completeness for each variable was computed as well as the mean completeness (with 95% CI) of filling the form. Completeness was further categorised as poor, average or good if the form had less than 50%, between 50%–75% and above 75% respectively of the variables on the form completely filled. Comparisons of estimates before and after the intervention were done using $\chi^2$ test. Significant differences were estimated at p=0.10 due to the relatively short intervention period. Qualitative data were audio recorded and transcribed verbatim, and all Twi and Ga responses were translated into English. Content analysis was carried out by MA-C and a research assistant with expertise in qualitative data analysis for patterns and emerging themes related to the study objectives. Differences were resolved through discussion between MA-C and DA. Main themes that were identified and triangulation of the FGDs, non-participant observations and IDIs data form the basis for reporting on and interpreting study findings.

## Ethical approval

Permission was obtained from the Greater Accra Regional Health Directorate and the participating district health directorates as well as the heads of the selected facilities. Written informed consent, assuring participants' safety, privacy and confidentiality of data provided, was obtained from all participating women and health workers for all parts of the study.

## Patient and public involvement

Patients were indirectly involved in the design of this study. Previous aggregate service data of patients seeking care within the GHS and specifically in the districts involved in this study informed our design and operationalisation of the study.

Also, although the intervention package was proposed before the study, engagement of patients and health workers as part of baseline qualitative data collection informed our modification of and finalisation of the intervention.

Recruitment of participants into the study was done by health workers based on the inclusion criteria. District and regional health service workers and managers supported the study.

Results of this study will first be shared with health workers and managers within the study districts as well the Greater Accra region of the GHS, since the intervention focused mainly on health worker practices with respect to obstetric referrals and interfacility communication.

Second, since provider practices we studied affect outcome of obstetric care in the three districts, and some of our findings suggest the need to educate women about the usefulness of referrals and thus the need to comply with it, community durbars will be organised in the districts to share relevant findings with the women and relevant stakeholders within the population.

## RESULTS

A total of 753 obstetric referrals were registered in the three district hospitals over the 9-month period of the study from 23 facilities. The facilities included three hospitals, one polyclinic, eight health centres, eight CHPS compounds and two community clinics. Apart from one hospital and one clinic that were privately owned, the other facilities were government owned. There were 313 referrals during the baseline period and 440 during the invention period. Out of these, only 280 (37.8%) had referral notes. During the baseline period, districts A, B and C had 62, 212 and 39 obstetric referrals, respectively, with 30, 38 and 17 referral notes, respectively. In the intervention period, there were 96, 312 and 32 referrals with 65, 115 and 15 referral notes, respectively, for the three districts. The specific reason for 210 (75.0%) of obstetric referrals was stated as 'for further management'. Two hundred and forty-seven (88.2%) of the referral notes were written by staff midwives, and for 11 (3.9%) notes, the category of the referring health worker was not stated. Figure 2 depicts an improvement in the proportion of clients with referral notes from the three districts comparing the baseline and intervention periods.

Non-participant observations did not show any discrimination in provision of referral note between emergency and non-emergency referrals but revealed that most emergency referrals were associated with inter-facility communication about the referral. Providing reasons for provision of referral notes in FGDs and IDIs at baseline, some health workers at the referring facilities reported they always give referral notes to pregnant women before they leave the facility while others stated that they sometimes do not give referral notes, especially if the referral is during the antenatal period and is not for an emergency. However, health workers at the district hospitals reported that not all obstetric referrals come in with referral notes. This trend was similar during the intervention period, but health workers explained that sometimes referred patients refuse to show the referral notes given to them because they want a completely different review and opinion at the referral facility or they may have gone home and reported to the referral facility much later than expected. A midwife corroborated these points during one of the FGDs in the following statement:

> As she rightly said sometimes when you give them the referral letter alone, they throw it away. Most of them don't like it when you refer them to the hospital because they think that they are going to end up with a caesarean section. So, they will throw the referral letter away. So, I write a referral letter and I write also in the book (maternal health record book). (Midwife, lower level facility, intervention period FGD, district C)

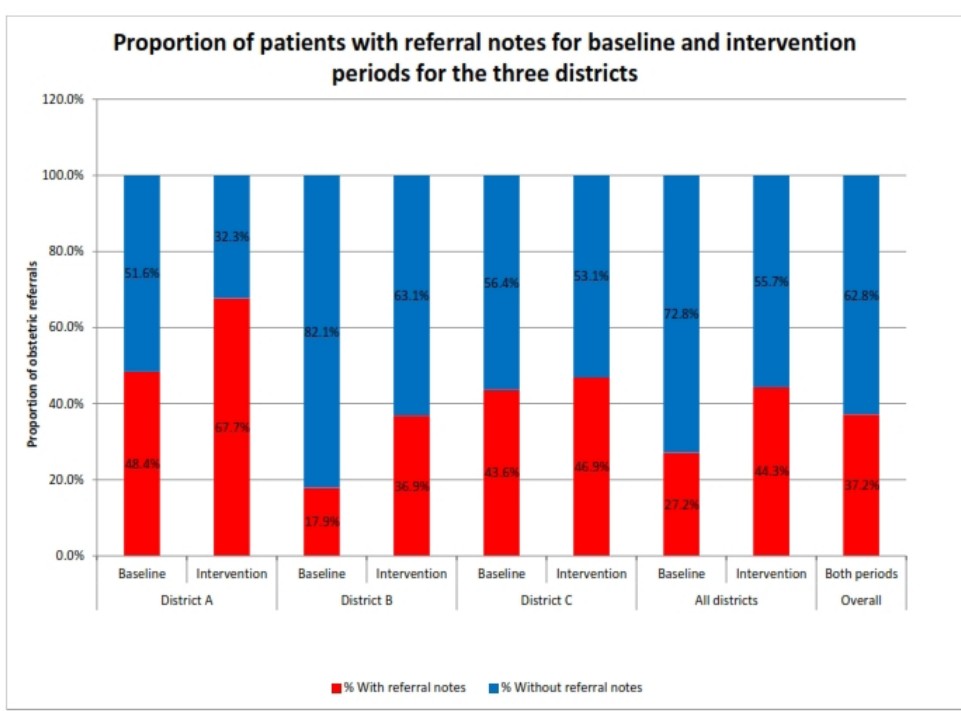

**Figure 2** A graph showing proportion of obstetric referrals with referral notes from the three districts, comparing baseline and intervention periods.

They also indicated that apart from the referral notes, they have referral notebooks at the facilities in which they keep record of all referrals.

In the FGDs with pregnant women, they reported that although they expect to be given referral notes during referrals, sometimes they are not given one. Some could not tell whether they were provided with one because one was not handed over to them with explanation of what it is for. They indicated that the referral noted is useful in sharing their information with the referring facility and that proves that they have indeed been referred.

… so if they can give us a note to send to the referral destination that is fine …… otherwise if they have the contact of the facility, they should call to tell them about the referred patient…… (Pregnant woman, baseline FGD, district A)

### Completeness of referral forms

During the baseline and intervention periods, 47.1% and 85.1%, respectively, of referral notes were written using the standard GHS referral forms. Other forms used

included printed facility adapted versions of the standard form (on which some variables were omitted), health insurance referral forms and prescription forms.[1] In few cases,[8] a summary of patient's notes was scribbled in the maternal health record book. For patient identification, 3.0% of notes did not record patient name, 2.0% did not record name of referring facility and 9.0% did not record patient age. There were variations in missing information on the forms for the clinical variables, patient complaints (22.0%), obstetric examination findings (22.0%), blood pressure (4.0%), diagnosis (2.0%) and management given (47.0%). Detailed information on completeness for each variable on the referral forms for baseline, intervention and overall study period is shown in table 2. Mean completeness of the referral forms (95% CI) for the three districts put together during baseline, intervention and overall period were 70.7% (60.4% to 80.9%), 71.9% (61.1% to 82.7%) and 71.3% (64.1% to 78.5%), respectively. When completeness was recategorised into poor, average and good, most of the forms (56.7%) were of average completeness (between 50.0% and 75.0%

**Table 2** Completeness of obstetric referral notes submitted to the three referral hospitals in three districts in the Greater Accra region, comparing baseline and intervention periods

| Category | Variable | Entered on form (n=85) Yes (N (%)) Baseline | Entered on form (n=195) Yes (N (%)) Intervention | Entered on form (n=280) Yes (N (%)) Overall | P value |
|---|---|---|---|---|---|
| Facility variables | Standard GHS referral form | 40 (47.1) | 166 (85.1) | 206 (73.6) | <0.01 |
| | Name and address of referring facility | 81 (95.3) | 193 (99.0) | 274 (97.9) | 0.05 |
| | Patient registration number | 26 (30.6) | 85 (43.6) | 111 (39.6) | 0.04 |
| | Time referred | 73 (85.9) | 181 (92.8) | 254 (90.7) | 0.07 |
| | Time patient left facility | 11 (12.9) | 25 (12.8) | 36 (12.9) | 0.98 |
| Patient identification | Patient name | 82 (96.5) | 190 (97.4) | 272 (97.1) | 0.66 |
| | Age | 76 (89.4) | 179 (91.8) | 255 (91.1) | 0.52 |
| | Patient insurance status | 54 (63.5) | 148 (75.9) | 202 (72.1) | 0.03 |
| Clinical variables | Patient complaints | 62 (72.9) | 155 (79.5) | 217 (77.5) | 0.23 |
| | Obstetric examination findings | 64 (75.3) | 155 (79.5) | 219 (78.2) | 0.43 |
| | Blood pressure | 83 (97.6) | 185 (94.9) | 268 (95.7) | 0.29 |
| | Weight | 47 (55.3) | 131 (67.2) | 178 (63.6) | 0.05 |
| | Laboratory findings | 37 (43.5) | 74 (37.9) | 111 (39.6) | 0.38 |
| | Diagnosis | 83 (97.6) | 192 (98.5) | 275 (98.2) | 0.64 |
| | Management given | 55 (64.7) | 92 (47.2) | 147 (52.5) | <0.01 |
| | Reason for referral | 75 (88.2) | 176 (90.3) | 252 (90.0) | 0.83 |
| | Position of referring officer | 77 (90.6) | 164 (84.4) | 240 (85.7) | 0.02 |
| | Signature of referring officer | 80 (94.1) | 188 (96.4) | 268 (95.5) | <0.01 |
| | Phone number of referring officer | 35 (41.2) | 39 (20.0) | 74 (26.4) | <0.01 |
| Completeness categorised | Poor | 3 (3.5) | 1 (0.5) | 4 (1.4) | 0.14 |
| | Average | 46 (54.1) | 113 (57.9) | 159 (56.7) | |
| | Good | 36 (42.4) | 81 (41.5) | 117 (41.8) | |
| | Mean completeness % (95% CI) | 70.67 (60.43 to 80.90) | 71.87 (61.10 to 82.65) | 71.31 (64.14 to 78.48) | 0.87 |

GHS, Ghana Health Service.

**Table 3** Comparison of completeness of referral notes among three districts in Greater Accra region

| Period | Variable | District A N (%) | District B N (%) | District C N (%) | P value |
|---|---|---|---|---|---|
| Baseline | Total referral notes | n=30 | n=38 | n=17 | <0.01 |
| | Poor completeness | 1 (3.3) | 1 (2.6) | 1 (5.9) | |
| | Average completeness | 9 (30.0) | 32 (84.2) | 5 (13.2) | |
| | Good completeness | 20 (66.7) | 5 (13.2) | 11 (64.7) | |
| Intervention | Total referral notes | n=65 | n=115 | n=15 | |
| | Poor completeness | 1 (1.5) | 0 (0.0) | 0 (0.0) | |
| | Average completeness | 42 (64.1) | 66 (57.4) | 5 (33.3) | |
| | Good completeness | 22 (33.8) | 49 (43.6) | 10 (66.7) | |
| Overall period | Total referral notes | n=95 | n=153 | n=32 | 0.01 |
| | Poor completeness | 2 (2.1) | 1 (0.7) | 1 (3.1) | |
| | Average completeness | 51 (58.7) | 98 (64.1) | 10 (31.3) | |
| | Good completeness | 42 (44.2) | 54 (35.3) | 21 (65.6) | |
| Overall period | Mean completeness (95% CI) | 68.99 (58.13 to 79.86) | 71.81 (60.50 to 83.12) | 77.34 (67.59 to 87.08) | 0.20 |

completely filled). There was no significant association between referring health worker category and category of completeness of the forms. Table 2 shows significant changes in completeness for only a few variables comparing the baseline and intervention periods. Overall, there was no significant change in mean completeness of forms from baseline to intervention period. In terms of category of completeness, there was a significant difference in the performance of the three districts (p=0.01), but the mean completeness showed no significant difference across the districts as shown in table 3 where the performance in the three districts are compared.

Exploring the reasons for incomplete referral forms, health workers and managers indicated during the FGDs and IDIs that, for medicolegal reasons, the referral forms are very important and need to be filled out completely and accurately and also serves as a guide to the health worker as to the essential details to share with the receiving facility during a referral. Incompletely filled forms make it difficult to manage the patient as one is not sure what had already been done for the patient, especially with medication. They however admitted that sometimes the referral notes are not completely filled, and this is a challenge for continuing care. They explained that they sometimes they do not fill the form completely because the patient's condition is serious and filling the form can be time wasting.

> Sometimes I do not fill it completely because the patient is in critical condition and has to be moved quickly to the next level. (Midwife, lower level facility, intervention period FGD, district B)

They also stated that most of the information is in the patient's maternal health record book so they find filling the referral form a duplication of effort. Another reason they attributed to not filling in some of the variables, like diagnosis, is that sometimes workers at the referral centre criticise them for referring patients with some specific

diagnosis. This embarrasses them so they rather leave the diagnosis blank. It is also for similar reasons that they indicate reason for referral in most of the notes as 'for further management'. Below is a midwife's account:

> Please when we refer the case with a referral letter they [health workers at referral center] should not make comments such as 'What is this, this case too you can't manage?' It happened when I accompanied a referred patient. I felt bad though I didn't write the referral letter and wondered if that is what goes on whenever we refer cases to bigger facilities. That practice is not professional and must stop. (Midwife, lower level facility, intervention period FGD, district A)

They also believe that any missing information on the referral form can be checked from the maternal health record book which the woman has in her possession.

### Availability of forms for referral notes

The standard GHS referral forms are procured from the regional medical stores. The facility audit at baseline showed that 19 out of 22 health facilities (86.4%) had the standard referral form booklets in stock. All hospitals, the one polyclinic and two clinics had the referral booklets in stock, with 87.6% and 75.0%, respectively, of health centres and CHPS compounds having them in stock. In the FGDs and IDIs during baseline, health workers and their managers reported that when they have stock-outs, they use photocopies of the forms. They fill two forms or use carbon to duplicate the filled form in order to get a second copy to keep at the facility as required. During the intervention period, a lot of facilities reported having run out of them and so used photocopied versions. When shown different or adapted versions of the standard forms that they had used over the study period, some health workers did not know that they were variants of the standard form without some of the required variables. Some reported that sometimes they write a summary of

the indication for referral in the maternal health record book because they do not think the patient's condition warrants a referral note or they had run out of stock.

## DISCUSSION
### Main findings
Only 37.8% of obstetric referrals from the three districts during the entire study period had referral notes. Provision of referral notes improved from 27.2% to 44.3%, respectively, from the baseline to the intervention period. For these notes, most (73.6%) were written using the standard GHS referral forms (47.1% and 85.1%, respectively, during the baseline and intervention period). Completeness of most forms was within the average category with mean completeness of 71.3% (64.1% to 78.5%) for the study period. During the FGDs and IDIs, health workers explained that they mostly write referral notes for emergency referrals and that most referral notes are not completely filled because other related information is in the maternal health record book which the women carry along to any facility.

### Strength and limitations
Strength of this study is the fact that we used both quantitative and qualitative methods, and this allowed us to both triangulate and explain the findings from the perspective of the health worker who refers patients and writes the notes.

Patients and referral notes were assessed at the referral hospital only (outside the referring facility). We were thus unable to assess referring provider and contextual factors associated with the completeness of the referral notes. This is a limitation of the study though the FGDs and IDIs helped us to minimise its effect. Another limitation is the fact that resources, including time, allowed us to implement and evaluate the intervention for a relatively short period than we would have desired. Considering the fact that providers provided more referral notes during the intervention period, it is a possibility that over time, with the intervention in place, mean completeness of the notes may have significantly improved as well.

### Implications for obstetric outcomes
Specialists who receive timely patient referral information are more likely to provide optimal care compared with those who do not.[17] Obstetric complications can be life threatening, and referrals of emergency obstetric cases without referral notes can be potentially time wasting for the receiving care provider.[18] Healthcare is a continuum, but with no prior information about a referred patient, the whole process of clinical management will have to start from scratch and that is undesirable when the patient needs urgent care. Although health workers gave the impression that all emergency patients get a note, this cannot be confirmed by the available data. The practice of not providing notes for non-emergency ANC referrals should not be encouraged either. Although the

maternal health record book contains ANC information for the woman, studies in Accra, Ghana, have shown gaps in ANC data both in the aggregate data and individual client record.[19 20] Regarding completeness of notes, incomplete information on medication, for example, is a serious concern. For example, there are implications for a woman with severe pre-eclampsia who has been given a loading dose of the medicine magnesium sulfate (MgSO4) before referral but for which no information exists on reaching the referral centre. Does the dose get repeated at the referral centre or not? How does this decision taken affect outcome for the patient? While MgSO4 toxicity or overload has grave consequences that can complicate the management of the patient,[21] the lack of the loading dose also puts the patient at a high risk of more seizures that worsen outcomes. This dilemma is avoided when the information is adequately provided on the referral form.

### Addressing challenges
Referral notes are very important component of the referral process. The desire is to have all referrals going out with a referral note as reported in one study.[6] Unfortunately, that was not always the case in our study. The use of standard referral forms or templates has been largely documented to improve documentation of important referral information as well as the overall quality of referral process.[22] There is therefore the need to continuously promote the use of the standard GHS referral form among providers of care. There were reported stock-outs of the standard referral forms, necessitating photocopying sometimes for use. This perhaps contributed to some referred patients, especially non-emergency and ANC clients, not getting referral notes. The stock-outs should be addressed with proper stock management of the booklets in the facilities. The referral teams that were formed were tasked to facilitate this role, and during the intervention period, utilisation of the standard form increased. Some clinicians however have expressed preference to rather use their own words to write referrals instead of using a standard form,[23] but reviews of such practice show letters with varying gaps.[24–26]

Training and supervision with feedback occurring alone or together have been shown as interventions that improve health worker performance especially in lower resource settings.[27–32] Specifically, studies have looked at the benefits of training on how to write referral notes[33] and the use of standard templates to improve quality of such notes from different categories of health workers. During the training in our study, health workers were reminded of the need to use the standard form and taken through the process of filling them out accurately. This together with feedback from the referral teams monthly review meetings on referrals contributed to the increase in provision of referral notes as well as use of the standard referral forms from 47.1% to 85.1% over the period as well. Such interventions that provide regular updates and feedback for health workers should be continued and possibly incorporated into

routine facility meetings and engagements. Time required for writing referral notes has been discussed as a problem by health workers,[9] but an understanding of the purpose a good referral note serves will help providers take up the task in an efficient manner.

It is important to counsel pregnant women during the ANC period on the importance of referrals and the need to pass on referral notes to the receiving facility so that they desist from hiding the notes. Some referred patients do not show up at the referral facility as has been reported in one study.[5] The patients expect clear communication and explanation of the diagnosis or indication for referral, treatment options and follow-up requirements during the referral process[9] and where this is lacking they have a challenge complying with instructions for referrals, including passing on referral notes. Referral facilities should also provide feedback to lower level facilities when their patients come without referral notes or incompletely filled referral notes so that these can be rectified in future.

Completeness of referral notes as shown in this study needs improvement. A study that looked at the content of referral notes or letters, although not specific to obstetric referrals, showed that over 90% of both generalists and consultants agreed that statement of the problem, current medication and reason for referral should be in a referral letter.[23] However, several studies show that referral notes from general practitioners often lack critical information such as reason for consultations, sociopsychological factors or plans for follow-up.[4 12 24–26] For example, one study found that although referring physicians provided patient background in 98.0% of referrals, they made the purpose of the referral explicit in only 76.0%.[12] Other studies showed no or very little information on physical examination and laboratory investigation on the referral letter.[34 35] The lack of adequate information has posed a challenge in using referral letters as tools for medical education,[26 36] although practitioners agree that referral notes should be used in professional audits.[23] Structured referral forms perform better with respect to completeness of information.[14] The referral protocol for GHS specifies that all the variables on the form must be filled in. This leaves no room for the health worker to use his or her discretion as to what variable to fill in and what not to for each patient. Although one study[33] showed that a letter with formatted content has the potential to enhance the quality of referral letters, other studies[23 37] showed that general practitioners preferred to have referral forms with less required items and would rather write a summary based on what they consider important for each patient rather than fill a form with mandatory fields. In that same study in Australia,[33] eight items were rated as essential information by a majority of referral letter recipients who are practitioners, and these include the diagnosis, clinical findings, test results, treatment options and recommendations, and prognosis. However, information pertaining to medical history, drug or social history was considered less essential. Some practitioners believe that the patient's characteristics as well as the circumstances of each case may vary the information that is essential in each referral note.[38] An assessment of the perspectives of the practitioners and the managers who designed the referral form as well as those who use them in Ghana will thus be important to appreciate how much of incompleteness is tolerable within the scope of obstetric referrals.

Electronic records have been shown to improve data completeness.[19 39–41] Few studies have examined the effects of electronic medical records on care coordination in general or on the referral process in particular.[42] Computer access to chart notes was associated with increased communication between referring physicians and specialists, with specialists receiving written or email referral letters more than twice as often as by telephone or other verbal communication.[8 17] Benefits of such electronic communication about referrals include the option for asynchronous communication and opportunities for back-and forth interchange and enhanced rapport.[43] Electronic notes are of better quality and also very useful and preferred by practitioners, especially if decision support functions are embedded in them.[29 33] Consideration can be given to linking them to essential health system components such as health insurance claims, with mandatory fields that cannot be skipped to optimise completeness of records. In another resource-rich setting, in the field of neurosurgery though, an online referral system was tested, and health workers found it very useful in communication and completing required documentation.[44] Employing their use in the Ghanaian context will be beneficial to the health system in general and referrals specifically.

## CONCLUSION

Referral notes were not provided for most obstetric referrals. The few referral notes were not always completely filled. Interventions such as training of health workers and regular review of referral processes can help improve both the provision and completeness of the referral notes. Use of electronic records should also be explored to benefit from its strengths.

**Author affiliations**
[1]Department of Epidemiology, University of Ghana, Noguchi Memorial Institute for Medical Research, Accra, Ghana
[2]Julius Center for Health Sciences and Primary Care, University Medical Center, Utrecht University, Utrecht, The Netherlands
[3]Center for Malaria Research, University of Health and Allied Sciences, Ho, Volta Region, Ghana
[4]Department of Biostatistics and Epidemiology, School of Public Health, Wits University, Johannesburg-Braamfontein, South Africa

**Acknowledgements** The authors are grateful to the Noguchi Memorial Institute for Medical Research (NMIMR) Postdoctoral Implementation Team for their support as well as the managers and workers of the Ghana Health Service (GHS) for supporting the process of implementing the intervention package and data collection. They are particularly grateful to Professor Kwadwo Koram for his guidance through the process.

**Contributors** MA-C conceived, designed and performed the study, analysed the data and wrote the paper. EA and KK-G contributed to the design of the study and

reviewed and approved the final version of the paper. DA contributed to the design of the study, analysed the data and reviewed and approved the final version of the paper.

**Funding** This study was funded by the WHO/TDR Postdoctoral grant number B40347 to the NMIMR.

**Competing interests** None declared.

**Patient consent for publication** Not required.

**Ethics approval** The entire study was approved by the NMIMR Scientific and Technical Committee and the Institutional Review Board (NMIMR-IRB CPN 072/16–17) as well as the GHS Ethical Review Committee (GHS-ERC:11/01/2017).

**Provenance and peer review** Not commissioned; externally peer reviewed.

**Data availability statement** Data are available on reasonable request.

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
