## [Reviewer comments · BMJ Open]

ARTICLE DETAILS

TITLE (PROVISIONAL)	Completeness of obstetric referral letters/ notes from sub-district to district level in three rural districts in Greater Accra region of Ghana: an implementation research using mixed methods
AUTHORS	AMOAKOH-COLEMAN, MARY; Ansah, Evelyn; Klipstein-Grobusch, Kerstin; Arhinful, Daniel

VERSION 1 – REVIEW

REVIEWER	Karen Finnegan PIVOT, Madagascar and United States Harvard Medical School, United States
REVIEW RETURNED	25-Feb-2019

GENERAL COMMENTS	Review the paper for editing. There are some incomplete sentences that should be modified. Also, you are inconsistent with your use of referring facility vs primary level facility; choosing one will make the paper clearer. Study design: Please quantify the intervention as much as possible. For example, in the first point about training, you could report how many people were trained, who was trained, and how long the training was. As the intervention works on multiple levels of the system, a figure may make things clearer. Methods: You document the percentage of women referred who have a completed referral form, but it's not clear how you know these women have been referred if data collection is occurring at the referral facility. You should also acknowledge as a limitation that you are unable to document the referral forms given to women who never complete their referral. Expand the description of the qualitative analysis. Who completed the content analysis? Was coding done by more than one person and how were differences in coding resolved? Related, you discuss FGDs, IDIs, and non-participant observation, but only share quotes from FGDs. Did the other forms of data collection inform your analysis? Ethics: Please be specific if you obtained written consent from women and health care workers for all parts of the study or just for participation in the qualitative data collection. Statistics: Please discuss what statistical testing you do to generate the p-values in Tables 2 and 3. Results: You have nearly complete data (according to Table 2) on the name of the referring facility. Are you able to look at
---

	differences in completeness by referring facility or is the concern that the numbers are too small? Regardless, the results would be strengthened by an understanding of the mean/median/range of number of referrals from each facility in baseline and the intervention period. In Table 3, you refer to Districts A, B, and C; previously you have been referring to them by name. Is there a reason for the change? If not, use district name consistently. Limitations: You don't note that one district has many more referrals than the others and, therefore, drives the results, especially at baseline. Also, the number of referrals increases quite a bit from baseline to the intervention period. Is there a way to review historic referral data and determine if this pattern is consistent with previous years? Both of these should be discussed in the limitations section if they are not able to be addressed with further analysis. Discussion: I was surprised by the introduction of electronic medical records in the discussion section, since it is a solution which seems unable to address some of the health workers concerns that you heard in focus groups, namely fear that the referral facility will shame them for not being able to do something and that in emergent situations there isn't time to complete a paper referral form. Is there literature you can reference on the use of electronic medical records in addressing these challenges?
--	--

REVIEWER	Prof Mattijs Numans MD PhD Dept Public Health and Primary Care, Campus The Hague, Leiden University Medical Centre, the Netherlands
REVIEW RETURNED	18-Mar-2019

GENERAL COMMENTS	An interesting mixed methods study on adequacy of referral documentation in a third world country. I have no major methodological concerns. The aim of the study is mainly to delineate possible improvement of referral procedures and letters and it comes up with a couple of interesting findings that have consequences for clinical practice in this kind of countries. The qualitative part indeed adds valuable information. The only suggestion would be to try to shorten the text, although this would not be very easy. However, I believe shortening to 3500 words would not cause essential information loss, when this is mainly done in the qualitative / focus group report.
---

REVIEWER	ENOCH ODAME ANTO Edith Cowan University School of Medical and Health Sciences 270 Joondalup Drive, Perth Australia
REVIEW RETURNED	13-Jun-2019

GENERAL COMMENTS	Manuscript ID bmjopen-2019-029785 Title: Completeness of obstetric referral letters/ notes from subdistrict to district level in three rural districts in Greater Accra region of Ghana: an implementation research General comments
--

	Authors have extensively done a good work and the manuscript is well-written. There are only minor corrections. References Reference numbers 2, 3 and 16 are incomplete. Authors must correct.
--	--

VERSION 1 – AUTHOR RESPONSE

Reviewer: 1

Reviewer Name: Karen Finnegan

Institution and Country: PIVOT, Madagascar and United States

Harvard Medical School, United States

Review the paper for editing. There are some incomplete sentences that should be modified. Also, you are inconsistent with your use of referring facility vs primary level facility; choosing one will make the paper clearer.

Response

The paper has been proof read.

The use of primary level facility is indeed deceptive since in Ghana, primary level facility refers to the district hospital and lower level facilities. The appropriate terminology therefore is either referring facility or lower level facility and this has been revised in the text appropriately.

Study design: Please quantify the intervention as much as possible. For example, in the first point about training, you could report how many people were trained, who was trained, and how long the training was. As the intervention works on multiple levels of the system, a figure may make things clearer.

Response

A figure has been provided to depict how the Intervention package was rolled out. Thanks

Methods: You document the percentage of women referred who have a completed referral form, but it's not clear how you know these women have been referred if data collection is occurring at the referral facility. You should also acknowledge as a limitation that you are unable to document the referral forms given to women who never complete their referral.

Response

We acknowledge that some women are referred but do not end up at the referral facility and this is supported by the qualitative data. However, our study participants for this paper are those who ended up at the referral facility (in this case the three hospitals). Out of these we assessed whether or not they arrived with a referral form. Our text reflects this under participants and quantitative method descriptions.

We have however added as a limitation the fact that it is possible that some participants may not have presented their referral notes as evident from our qualitative results.

In another manuscript related to this study which we are working on, we share monthly aggregate statistics of all referrals from lower levels of care, including those that do not end up in hospitals.

Expand the description of the qualitative analysis. Who completed the content analysis? Was coding done by more than one person and how were differences in coding resolved? Related, you discuss FGDs, IDIs, and non-participant observation, but only share quotes from FGDs. Did the other forms of data collection inform your analysis?

Response

We have provided more clarity on the qualitative analysis as suggested (lines 206-211).

Although we provide the quotes from the FGDs, (because we believe they make the points stronger) we have mentioned under results that the qualitative results are drawn from both FGDs and IDIs and make references to that in the text. The challenge with word count made us go for the most compelling statements made by health workers that buttress the point being reported. We do have some compelling statements in the IDIs but not related for the focus of this paper of referral forms.

We have added a statement on relevant information from the non-participant observations (lines 257-259)

Ethics: Please be specific if you obtained written consent from women and health care workers for all parts of the study or just for participation in the qualitative data collection

Response

Written informed consent was obtained from all participants in the study, which includes both quantitative and qualitative parts. We have clarified that in the text under ethics by adding "for all parts of the study" (line 220).

Statistics: Please discuss what statistical testing you do to generate the p-values in Tables 2 and 3.

Response

We used chi-squared test. The text has been revised to reflect that (line 204).

Results: You have nearly complete data (according to Table 2) on the name of the referring facility. Are you able to look at differences in completeness by referring facility or is the concern that the numbers are too small? Regardless, the results would be strengthened by an understanding of the mean/median/range of number of referrals from each facility in baseline and the intervention period.

Response

Yes, this will be in small numbers considering that we have referrals coming from 16 lower level facilities and the total numbers of referral forms are generally small. We however appreciate that dimension and what value it would have added to the paper.

In Table 3, you refer to Districts A, B, and C; previously you have been referring to them by name. Is there a reason for the change? If not, use district name consistently.

Response:

We have used District A, B and C under results for anonymity and confidentiality during publication. In dissemination of findings locally these pseudo names will be dropped. For consistency, we have revised and used Districts A, B and C under methods now.

Limitations: You don't note that one district has many more referrals than the others and, therefore, drives the results, especially at baseline. Also, the number of referrals increases quite a bit from baseline to the intervention period. Is there a way to review historic referral data and determine if this

pattern is consistent with previous years? Both of these should be discussed in the limitations section if they are not able to be addressed with further analysis.

Response

Although one district has more referrals (because it is peri-urban with a higher population as mentioned under settings), the analysis for this paper focuses on proportion of these referrals with referral notes which reflects practice. Thus, we do not think this is a limitation. In further analysis of the data in subsequent papers, we will do the required corrections for the increased numbers from that one district.

Obstetric data in Ghana, unlike malaria for example, has not been shown to be seasonal or follow a pattern and so we do not believe the change in numbers between intervention and baseline warrants sub-analysis. The numbers actually dropped for one district out of the three from baseline to intervention period.

Discussion: I was surprised by the introduction of electronic medical records in the discussion section, since it is a solution which seems unable to address some of the health workers concerns that you heard in focus groups, namely fear that the referral facility will shame them for not being able to do something and that in emergent situations there isn't time to complete a paper referral form. Is there literature you can reference on the use of electronic medical records in addressing these challenges?

Response

Your perspective on this point are very much appreciated. We draw on the evidence shown in the references we cited to support the introduction of electronic records to improve provision and completeness of referral forms. We believe as one of the papers mention that if decision support functions are embedded in electronic records, and also for example linked to health insurance claim processing for those who are insured, providers will find it important enough to fill the forms for patients since it will be tied in to their revenue. We however do not propose that it will solve all the challenges that make providers not provide completely filled forms. Issues of 'shaming" for example have to be dealt with through other avenues such as health worker capacity building and reorientation.

Response

Reviewer: 2

Reviewer Name: Prof Mattijs Numans MD PhD

Institution and Country: Dept Public Health and Primary Care, Campus The Hague, Leiden University Medical Centre, the Netherlands

An interesting mixed methods study on adequacy of referral documentation in a third world country. I have no major methodological concerns. The aim of the study is mainly to delineate possible improvement of referral procedures and letters and it comes up with a couple of interesting findings that have consequences for clinical practice in this kind of countries. The qualitative part indeed adds valuable information. The only suggestion would be to try to shorten the text, although this would not be very easy. However, I believe shortening to 3500 words would not cause essential information loss, when this is mainly done in the qualitative / focus group report.

Response

Thanks for the feedback. Shortening the text is very desirable for us and various revisions to reduce the material took place before this version. Our challenge in reducing the text further lie in the fact that:

1. This is the first paper under review for publication for the entire project so we need to be elaborate on the methods, which are both quantitative and qualitative, as well as describe the intervention package in detail.
 2. The presentation of the results to include the qualitative aspect of our work, which strengthens the contextual issues, also adds to the text significantly.
- We are grateful the editorial policy of BMJ Open allows us to put in this much.

Reviewer: 3

Reviewer Name: ENOCH ODAME ANTO

Institution and Country: Edith Cowan University, School of Medical and Health Sciences, Perth, Australia

Manuscript ID bmjopen-2019-029785

Title: Completeness of obstetric referral letters/ notes from subdistrict to district level in three rural districts in Greater Accra region of Ghana: an implementation research

General comments

Authors have extensively done a good work and the manuscript is well-written. There are only minor corrections.

References

Reference numbers 2, 3 and 16 are incomplete. Authors must correct.

Response

Thank you for your feedback. Manuscript has been proof read.

References have the rechecked and revised manually since the referencing software kept omitting relevant information.

References 2 and 16 are unpublished Master's Theses and 3 is a Report.

Reference 3 is a Report.

VERSION 2 – REVIEW

REVIEWER	Karen Finnegan Harvard Medical School United States of America
REVIEW RETURNED	06-Aug-2019

GENERAL COMMENTS	Thank you for this interesting study. You've combined the quantitative and qualitative data to generate interesting findings. My only suggestion is that the district names should be removed from the graph.
--